

# Multiple factors influence local perceptions of snow leopards and Himalayan wolves in the central Himalayas, Nepal

Madhu Chetri[1], Morten Odden[1], Olivier Devineau[1], Thomas McCarthy[2] and Per Wegge[3]

[1] Faculty of Applied Ecology, Agricultural Sciences and Biotechnology, Inland Norway University of Applied Sciences, Evenstad, Norway
[2] Panthera, New York, NY, USA
[3] Faculty of Environmental Sciences and Natural Resource Management, Norwegian University of Life Sciences, Ås, Norway

## ABSTRACT

An understanding of local perceptions of carnivores is important for conservation and management planning. In the central Himalayas, Nepal, we interviewed 428 individuals from 85 settlements using a semi-structured questionnaire to quantitatively assess local perceptions and tolerance of snow leopards and wolves. We used generalized linear mixed effect models to assess influential factors, and found that tolerance of snow leopards was much higher than of wolves. Interestingly, having experienced livestock losses had a minor impact on perceptions of the carnivores. Occupation of the respondents had a strong effect on perceptions of snow leopards but not of wolves. Literacy and age had weak impacts on snow leopard perceptions, but the interaction among these terms showed a marked effect, that is, being illiterate had a more marked negative impact among older respondents. Among the various factors affecting perceptions of wolves, numbers of livestock owned and gender were the most important predictors. People with larger livestock herds were more negative towards wolves. In terms of gender, males were more positive to wolves than females, but no such pattern was observed for snow leopards. People's negative perceptions towards wolves were also related to the remoteness of the villages. Factors affecting people's perceptions could not be generalized for the two species, and thus need to be addressed separately. We suggest future conservation projects and programs should prioritize remote settlements.

## INTRODUCTION

Large carnivore co-existence with humans remains a global challenge (*Athreya et al., 2013*), and mitigation of human-carnivore conflicts requires multiple approaches and disciplines (*Redpath et al., 2013*). Among the various aspects of carnivore conflict management, understanding local perceptions is crucial for establishing long term conservation strategies (*Bagchi & Mishra, 2006*; *Conforti & Cesar Cascelli de Azevedo, 2003*), especially in multi-use landscapes where animal husbandry is the main source of income. An assessment of local perceptions helps in identifying groups of people or

Corresponding author
Madhu Chetri, madhu.chetri@inn.no

villages that are negative towards protection of carnivores, and thus aids conservation authorities to find suitable strategies to improve their tolerance (*Suryawanshi, 2013*; *Treves & Karanth, 2003*). Further, assessments form a basis for quantifying the effects of conservation management interventions and aid in formulating new strategies if opinions towards conservation change (*Dressel, Sandström & Ericsson, 2015*).

Globally, local perceptions and attitudes towards large carnivores are complex and vary markedly between regions (*Røskaft et al., 2007*). Multiple factors influence local perceptions, including animal behavior, risk of negative encounters, and the length of the period of co-existence (*Dickman, 2010*; *Dressel, Sandström & Ericsson, 2015*; *Kellert et al., 1996*; *Zimmermann, Wabakken & Dötterer, 2001*). Local perceptions also vary among ethnic groups, and are linked to religious and cultural beliefs (*Ale, Shah & Jackson, 2016*; *Dickman et al., 2014*; *Kellert et al., 1996*; *Li et al., 2013*; *Mkonyi et al., 2017*). Socio-demographic variables such as age, sex, income, occupation, literacy, number of livestock owned and loss to predators have all shown to be associated with local perceptions and attitudes towards large carnivores (*Caruso et al., 2020*; *Fort et al., 2018*; *Kellert & Berry, 1987*; *Kideghesho, Røskaft & Kaltenborn, 2007*; *Naughton-Treves, Grossberg & Treves, 2003*; *Oli, Taylor & Rogers, 1994*; *Røskaft et al., 2007*; *Trajçe et al., 2019*).

In the central Himalayas, Snow leopards (*Panthera uncia*) and Himalayan wolves (*Canis lupus chanco*) are the two most important predators involved in conflicts with people (*Chetri et al., 2019a*). A recent study from the region revealed that snow leopards were responsible for the majority of predation losses (61.9%); the remaining were from Himalayan wolf (16.8%) and other predators (21.3%) including feral dogs (*Canis lupus familiaris*), brown bear (*Ursus arctos*), black bear (*Ursus thibetanus*), Eurasian lynx (*Lynx lynx*), golden jackal (*Canis aureus*) and common leopard (*Panthera pardus*) (*Chetri et al., 2019a*). The snow leopard is categorized as Vulnerable in the IUCN red list of threatened species (*McCarthy et al., 2017*), whereas wolf is considered as Least Concern (*Boitani, Phillips & Jhala, 2018*). However, in the national Red Data List of Nepal, wolves are considered as Critically Endangered and snow leopards are considered as Endangered (*Jnawali et al., 2011*). A recent fecal DNA study reported that the snow leopard density within our study area in the central Himalayas was 0.95 (SE 0.19) animals per 100 km$^2$ (*Chetri et al., 2019b*), but density estimates of wolves from the area are still lacking (*Chetri et al., 2016*; *Chetri, Odden & Wegge, 2017*). The species has received little conservation attention due to its lower conservation status in the IUCN Red list, which has made it difficult to acquire necessary funding for population monitoring. According to *Chetri (2014)*, Himalayan wolves are rare in the region and mostly solitary. The wolves that thrive in this landscape are genetically unique to the region as revealed by recent DNA analysis, and they are considered different from the gray wolf lineage (*Chetri et al., 2016*). Both species range widely and often encounter pastoralists.

Although information on livestock depredation by snow leopards and wolves exists from Nepal's Himalaya (*Aryal et al., 2014*; *Chetri, Odden & Wegge, 2017*; *Chetri et al., 2019a*; *Oli, Taylor & Rogers, 1994*; *Wegge, Shrestha & Flagstad, 2012*; *Werhahn et al., 2019*), limited information is available regarding variation in local perceptions and tolerance to these species on a large spatial scale (*Hanson, Schutgens & Leader-Williams, 2019*;

*Kusi et al., 2019*; *Oli, Taylor & Rogers, 1994*). Hence, in our study, we examined local communities' perceptions of snow leopards and wolves in a large area of ~5,000 km$^2$ where livestock depredation has been a main concern in recent decades. The survey covered two protected areas, Annapurna Conservation Area (ACA, hereafter) and Manaslu Conservation Area (MCA, hereafter), where ecological studies of snow leopard and wolf had recently been conducted (*Chetri, 2018*). These studies showed that snow leopard density was far lower than previously assumed, and consequently, average annual livestock losses were low (ca.1%) even though livestock constituted large proportions of the diet of both snow leopards and wolves (ca.25%). Despite the low levels of livestock depredation, perceptions of wolves are often negative. Similarly, incidents of mass killings of livestock by snow leopards decreases local tolerance towards their conservation, which in turn may lead to retaliatory killing of carnivores (*Jackson, 2015*; *Mishra, Redpath & Suryawanshi, 2016*; *Mishra et al., 2004*; *Suryawanshi et al., 2014*; *Woodroffe et al., 2005*). Surplus killings and injuries of high valued livestock (e.g., horses, milking yaks and cows) not only outrage local communities (*Oli, Taylor & Rogers, 1994*), but also have negative repercussions that can spread even to distant villages.

Integrated conservation and development efforts that were initiated in ACA and MCA in the 1990-ies included conservation awareness campaigns principally targeting snow leopard, but not wolves. Due to the relatively low livestock losses and the considerable conservation efforts in the study area, we expected perceptions of carnivores to be more positive than reported from previous studies from the Himalayan range. Furthermore, we expected perceptions to vary geographically as well as between species due to a bias in the impact of conservation awareness campaigns, that is, tolerance of wolves should be lower, and perceptions could potentially depend on the remoteness of villages. Lastly, we expected perceptions to be affected by socioeconomic and demographic factors, for example, livestock losses and ownership, gender, age, education and occupation.

## MATERIALS AND METHODS

### Ethics statement

Approval and relevant permits required to carry out this research were obtained from the National Trust for Nature Conservation (NTNC) (Ref.no. 291), Nepal.

### Study area

We conducted the study in the Annapurna–Manaslu Conservation landscape in the central Himalayas (N28–29°, E83–85°; Fig. 1). Both ACA and MCA are located within this landscape and are the largest community-based conservation areas in Nepal (ca.9,292 km$^2$). It is located in the rain shadow area of the Himalayas. Together with Bhimthang valley, it is the priority landscape for snow leopards conservation in the country (*DNPWC, 2017*). The human population density is 1 per km$^2$ (*CBS, 2012*), and agro-pastoralism is the main source of livelihood, although some households are also involved in eco-tourism related enterprises. The overall livestock density in the study area is 35.74 ± 0.10/km$^2$ (*Chetri, Odden & Wegge, 2017*). All accessible areas are used for livestock grazing following the seasonal traditional Tibetan calendar (*Chetri et al., 2019a*).
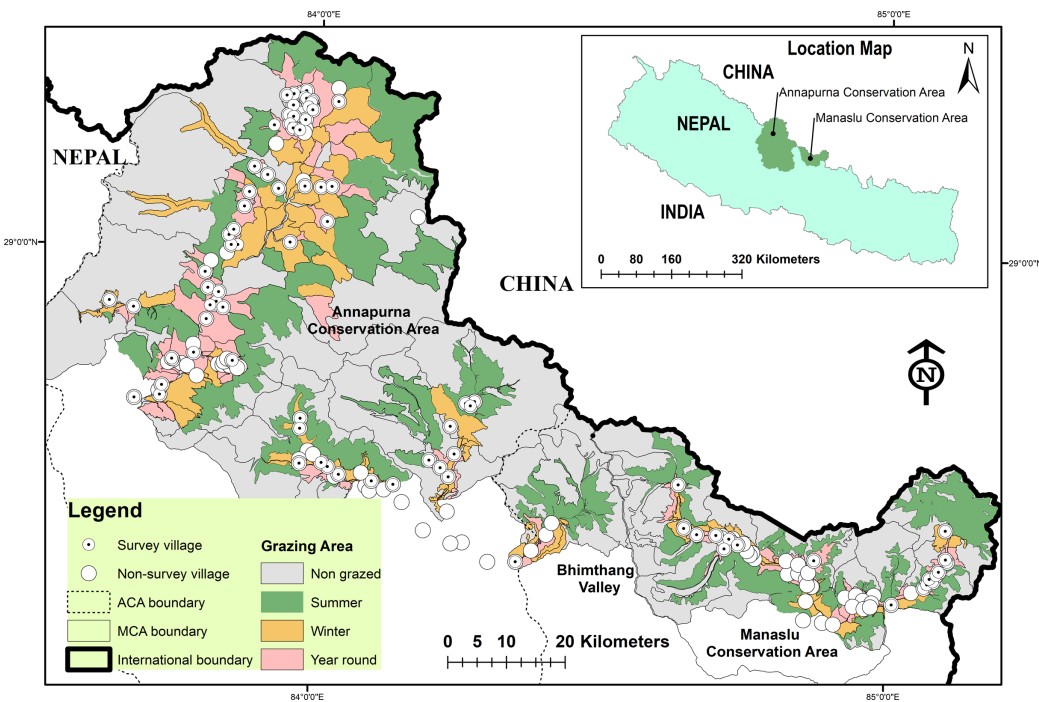

**Figure 1 Study area with location of survey villages and grazing areas.**

Grazing areas are designated for seasonal grazing as summer and winter. Areas close to villages are used for year-round grazing (Fig. 1). Livestock, for example sheep, goats and cows, are usually herded and periodically moved among different pastures according to seasons (*Chetri, Odden & Wegge, 2017*). Small stocks (sheep and goats) are herded and sometimes accompanied by herding dogs. They are released in the morning and brought back to corrals/pens in the afternoon on a regular basis. Similarly, milking cows and, yaks are brought back to corrals/pens in the afternoon or in the morning for milking. Livestock are kept in corrals/pens for protection against predators. Corrals are traditionally made of mud walls and stones.

Over the last decade there have been considerable changes in the lifestyle of the local people due to the development of roads in ACA. Despite these changes, traditional agro-pastoral lifestyles remain intact, and most importantly, traditional livestock grazing and collective village level decision making and implementation is still functional. In ACA, most farmers prefer to raise goats (*Capra hircus*) and sheep (*Ovis aries*) due to abundant pastureland, whereas in MCA farmers prefer cattle-yak hybrids (dzo, Jhopas, *Bos* spp.) as they are both grazers and browsers. Similarly, in the central part of ACA, farmers prefer yaks (*Bos grunniens*) due to dominant scrub vegetation. Lulu cows (*Bos taurus* sp.) and horses (*Equus ferus caballus*) are common in all areas. Among the main wild ungulates, bharal (*Pseudois nayaur*) and Himalayan tahr (*Hemitragus jemlahicus*) are widespread, whereas Tibetan argali (*Ovis ammon hogdsoni*), kiang (*Equus kiang*), and Tibetan gazelle (*Procapra picticaudata*) have overlapping grazing areas with livestock in the north-western parts of ACA.

Apart from snow leopards and wolves, other carnivores include golden jackal, red fox (*Vulpes vulpes*), Himalayan black bear, Tibetan sand fox (*Vulpes ferrilata*), brown bear, Eurasian lynx and several species of weasel (*Mustela* spp.), and marten (*Martes* spp.).

## Semi-structured questionnaire survey

To assess local perceptions and tolerance towards snow leopards and wolves, we administered a semi-structured questionnaire (see Supplemental Materials for questionnaire structure, Appendix S1). We randomly selected 428 households scattered over 85 settlements within 21 Village Development Committee (VDC) units from July to September 2014. A VDC is the lowest rural administrative unit and usually encompasses 7–9 small clustered villages or settlements know as a ward. Recently, with the promulgation of the new constitution, the Nepal government dissolved the VDC structure and established a new local body known as gaun palika or rural municipality. But in this study we will use VDC as the data was collected prior to this change.

Each VDC has separate designated grazing areas. Among the 21 VDCs in the study area (6,621 km$^2$), 2,934 km$^2$ (44.3%) was used for livestock grazing (summer 55.6%, winter 24.6% and 19.8% year-round). The remaining areas (ca.3,687 km$^2$) were inaccessible for livestock grazing due to rugged terrain and high altitude (Fig. 1). Our survey covered 13% of the total number of households within the survey villages (*CBS, 2012*). Due to scattered settlements/villages, vast landscape and remoteness of the area, most of the questionnaires were conducted using locally trained community members, managed through the Unit Conservation Offices (UCOs) of ACA and MCA (*Chetri et al., 2019a*). Before the initiation of the survey, each interviewer was briefed about the purpose of the study and trained in how to conduct the semi-structured questionnaire, and verbal consent was obtained from all subjects. The survey households were selected following the main village trails. We approached the household closest to the main village trail and selected every third household thereafter for interviews. If the inhabitants were absent, we selected the nearest neighbor. For each respondent, we recorded the number of livestock owned, herd composition and livestock loss to snow leopards and wolves during the previous year. We also recorded respondent age, gender, education and occupation. We asked their opinion about the presence of snow leopards and wolves near their grazing areas and homesteads, and categorized their answers as positive, neutral and negative. We considered questionnaires as invalid when respondents stated that they did not know about the species presence and conflict (i.e., neutral responses). These questionnaires were excluded from the analyses. Hence, although we administered similar questionnaire sets to assess perceptions of snow leopards and wolves, the sample size for wolf perceptions became smaller due to a larger proportion of invalid questionnaires (Table 1). This was mainly because wolves are found only in the northwestern section of ACA and MCA (see Fig. 1).

## Data analysis

We used generalized linear mixed effects models (GLMMs) to determine the relationship between response and potential predictor variables with two separate sets of models, one

**Table 1 Overview of respondent characteristics in the Annapurna–Manaslu landscape, central Himalayas, Nepal.**

| | | Snow leopards | Wolves |
|---|---|---|---|
| Number of questionnaires | | 428 | 428 |
| Number of included/valid questionnaires | | 395 (92.3) | 327 (76.4) |
| Number of invalid/excluded questionnaires | | 33 (7.7) | 101 (23.6) |
| Village Development Committee (VDC) | | 21 | 14 |
| Respondent age range (years) | | 20–87 | 20–87 |
| Mean age (years) | | 46 | 46 |
| Region | ACA | 353 (89.4) | 296 (90.5) |
| | MCA | 42 (10.6) | 31 (9.5) |
| Gender | Male | 331 (83.8) | 270 (82.6) |
| | Female | 64 (16.2) | 57 (17.4) |
| Education | Literate | 213 (53.9) | 180 (55.0) |
| | Illiterate | 182 (46.1) | 147 (45.0) |
| Occupation | Agro-pastoralist | 341 (86.3) | 291 (89.0) |
| | Livestock herding | 30 (7.6) | 13 (4.0) |
| | Others | 24 (6.1) | 23 (7.0) |
| Perceptions | Positive | 186 (47.1) | 51 (15.6) |
| | Negative | 209 (52.9) | 276 (84.4) |

Notes:
Only respondents that responded to perceptions and share their experiences were included. Numbers in parenthesis indicates percentage of individual respondents in each category.
ACA (hereafter), Annapurna Conservation Area.
MCA (hereafter), Manaslu Conservation Area.

for snow leopards and one for wolves. GLMMs take into account random effects and provide a more flexible approach for analyzing non-normal data (*Bolker et al., 2009*). As a binomial response variable, we categorized opinions of presence of snow leopards and wolves as either positive or negative, as described previously. As explanatory variables, we included factors and covariates that were identified as important predictors of livestock losses in a previous study conducted in the region (*Chetri et al., 2019a*), as well as demographic and socioeconomic variables that have been linked to perceptions of carnivores in previous studies (*Caruso et al., 2020*; *Kellert & Berry, 1987*; *Kusi et al., 2019*; *Oli, Taylor & Rogers, 1994*; *Suryawanshi et al., 2014*): (i) Ownership (number of livestock owned), (ii) herd composition (proportion of large stock relative to all livestock owned, see *Chetri et al. (2019a)*), (iii) number of livestock lost to snow leopards or wolves, (iv) occupation (agro-pastoral, livestock herding, and others, that is, respondents benefitting from tourism, such as traders, hotel owners, tourist guides and porters), (v) literacy (ability to read and write), (vi) gender, (vii) age, and (viii) distance to the nearest conservation office (number of walking days required to reach nearest conservation field office—standardized to 8 h walking/day). We did not include religion as a predictive factor as all the locals residing in the region are Buddhist, while only a few outsiders who are working as laborers or teachers are non-Buddhist. We standardized all numeric explanatory variables by two standard deviations, following *Gelman & Hill (2007)*.

VDC was used as a random effect in all models. We checked correlation among (continuous) predictors before carrying out the regression analysis, and we did not include collinear variables (rho > 0.6) into the same model. We analyzed the data using R version 3.4.2 (*R Core Team, 2017*).

## RESULTS

We used only 395 and 327 questionnaires in our analyses regarding snow leopards and wolves, respectively (Table 1), due to the exclusion of respondents that were neutral or unaware of species presence or conflicts. In terms of gender, more than 80% of the respondents were male (Table 1), and approximately 50% of the respondents were illiterate. Among occupations, most respondents belonged to the agro-pastoralist category (Table 1). An analysis of respondents' perceptions in general revealed that local people were more negative towards wolves ($n = 276$, 84.4%) than towards snow leopards ($n = 209$, 52.9%) (Table 1).

### Local perceptions of snow leopards

We compared 22 candidate models to assess perceptions of snow leopards (Table S1). The two highest ranking models had a small difference in AICc value (ΔAICc = 0.27) and Akaike weights (0.45 and 0.39). Both models included the predictor variables occupation, sex and the interaction between age and literacy (Table 2). The top ranking model in terms of AICc also included ownership, but due to the marginal effect of removing this variable, we present here the simpler second ranking model (Fig. 2). The occupation of the respondents had a strong effect on their perceptions of snow leopards. Among the three categories of occupation, there was only a slight difference in perceptions between agro-pastoralists (income from both agriculture and livestock) and herders (income solely from livestock herding). On the contrary, respondents with other additional sources of income (e.g., tourism) were far more positive towards snow leopards (i.e., OCCOTHER, Fig. 2). Furthermore, sex was included in the model, but the effect was weak, that is, men were more positive towards snow leopards than women. The main effects of literacy and age had very weak impacts on perceptions, but the interaction between these terms showed a marked effect. As illustrated in Fig. 2, being illiterate had a more marked negative impact among the older respondents.

### Local perceptions of wolves

We compared 24 candidate models to assess perceptions of wolves (Table S2). The highest ranking model performed far better than the other candidates (Akaike weight = 0.70, Table 2). This model included different predictors than the model for snow leopards, that is, sex and ownership (Fig. 3). In this case, male respondents were markedly more positive than females. Other predictor variables were livestock loss (numbers lost to wolves), herd composition, distance to the nearest conservation field office and livestock ownership. The latter predictor had a marked effect on perceptions, that is, respondents with larger herds were more negative.

**Table 2 Model selection for perception towards the snow leopard and the Himalayan wolf.**

| Model | df | logLik | AICc | delta | Weight |
|---|---|---|---|---|---|
| **Snow Leopard** | | | | | |
| OWN + OCC + SEX + AGE * LIT | 9 | −233.24 | 484.9 | 0 | 0.45 |
| OCC + SEX + AGE * LIT | 8 | −234.42 | 485.2 | 0.27 | 0.39 |
| COMP * LOSS + OWN + OCC + SEX + AGE * LIT | 12 | −231.64 | 488.1 | 3.15 | 0.09 |
| COMP * LOSS + OCC + SEX + AGE * LIT | 11 | −233.63 | 490 | 5.02 | 0.04 |
| COMP * LOSS + OWN + OCC + SEX + AGE | 10 | −234.74 | 490 | 5.11 | 0.04 |
| OCC | 4 | −245.06 | 498.2 | 13.28 | 0 |
| COMP * LOSS + OWN + OCC | 8 | −241.52 | 499.4 | 14.47 | 0 |
| COMP * LOSS + OWN + OCC + SEX | 9 | −240.97 | 500.4 | 15.46 | 0 |
| AGE | 3 | −247.76 | 501.6 | 16.64 | 0 |
| LIT | 3 | −251.48 | 509 | 24.08 | 0 |
| **Himalayan Wolf** | | | | | |
| COMP + DIST + LOSS + SEX + OWN | 7 | −104.38 | 223.1 | 0 | 0.7 |
| COMP * LOSS + OWN + OCC + SEX | 9 | −104.88 | 228.3 | 5.22 | 0.05 |
| SEX | 3 | −111.33 | 228.7 | 5.62 | 0.04 |
| OWN | 3 | −111.63 | 229.3 | 6.22 | 0.03 |
| COMP + LOSS + OWN | 5 | −109.59 | 229.4 | 6.26 | 0.03 |
| COMP + LOSS + OWN + DIST | 6 | −108.75 | 229.8 | 6.65 | 0.03 |
| COMP * LOSS + OWN + OCC + SEX + AGE | 10 | −104.86 | 230.4 | 7.31 | 0.02 |
| OWN + OCC + SEX + AGE * LIT | 9 | −106.05 | 230.7 | 7.56 | 0.02 |
| COMP + LOSS * OWN | 6 | −109.26 | 230.8 | 7.67 | 0.02 |
| COMP * LOSS + OWN + LIT | 7 | −108.3 | 230.9 | 7.83 | 0.01 |

**Notes:**
All continuous variables were standardized by 2 standard deviations (as per *Gelman & Hill, 2007*) and all models included a varying intercept on VDC (Village Development Committee). VDC is included as a random effect. AGE, age of the respondent; COMP, composition of the herd; that is, proportion of large stock animals, LIT, literacy (yes/no); LOSS, number of domestic animals lost to the carnivore; OCC, respondent's occupation (Herding, Agriculture, Other); OWN, number of domestic animals owned; SEX, gender of the respondent; DIST, Distance from the nearest conservation field office to respondent household. Only the top 10 models are presented for each analysis.

## DISCUSSION

In our study landscape, a far larger proportion of respondents were negative towards wolves than to snow leopards. This was also observed by *Kusi et al. (2019)* in upper Dolpa and Humla areas, located in the western region of Nepal. This perception is common in areas where wolves coexist with other large predators, for example brown bear and lynx (*Trajçe et al., 2019*). A possible cause is related to the difference in behavior of wolves compared to other carnivores. Snow leopards are cryptic, avoid humans and are more nocturnal than wolves (*McCarthy, Fuller & Munkhtsog, 2005*; *Mech & Boitani, 2010*). Wolves are more active during the day and attack livestock both during day and night (*Xu, Yang & Dou, 2015*). Furthermore, research has shown that greater visibility and howling behavior of wolves may reinforce negative perceptions (*Kellert et al., 1996*).

Social norms and cultural beliefs also play an important role in perceptions of the two carnivores. Cultural sentiments, religious belief and folklore associated with snow leopards have a strong positive influence on their conservation (*Ale, Shah & Jackson, 2016*;

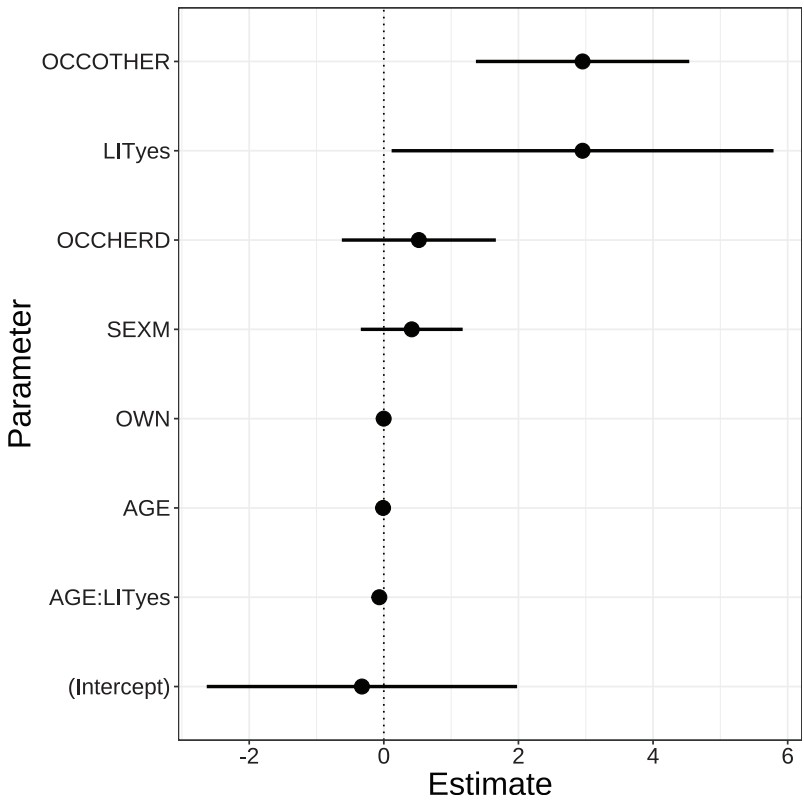

**Figure 2 Parameter estimates based on Generalized Linear Mixed-Effects Models of factors affecting perceptions of snow leopards.** Dots and solid lines represent parameter estimates and 95% CI. The estimates are on the logit scale. The strength of the effect of parameters is indicated by the distances between the solid horizontal lines and the dotted vertical line. VDC (Village Development Committee) is included as a random effect. OCCOTHER, Occupation-Others; LITyes, Literate (read and write); OCCHERD, Occupation-Herding; SEXM, Male; OWN, Total livestock holding; AGE, Age of the respondent; AGE:LITyes, the interaction between the respondent's age (AGE) and literacy (LITyes).

*Li et al., 2014*). Further, the local practice of non-violence (e.g., Tsum valley of MCA) and protection of forest and landscape in the name of monasteries (*Li et al., 2014*) have also played an important role in snow leopard conservation. The Buddhist communities to which most of our respondents belong traditionally do not kill wildlife because it was considered a sin in their religion (*Li et al., 2014*). The snow leopard is often considered as a symbol of the mountains, and the charisma of the species promotes attention both in terms of research and conservation efforts from global and national conservation authorities (*McCarthy et al., 2016*). In contrast, wolves are traditionally depicted as merciless and evil creatures in legends and folklore (*Dingwall, 2001*; *Marvin, 2012*). A recent study from Spiti, India showed that more than 98% of the survey respondents claimed that wolves were not safe for livestock and their presence was highly disliked by the communities (*Lyngdoh & Habib, 2019*). Similar trends have been observed in parts of Europe (*Dingwall, 2001*; *Marvin, 2012*) and America (*Grima, Brainard & Fisher, 2019*). This is not surprising as dislike to wolves is common across the globe (*Bhatia et al., 2016*; *Dressel, Sandström & Ericsson, 2015*; *Kansky, Kidd & Knight, 2014*; *Kusi et al., 2019*;

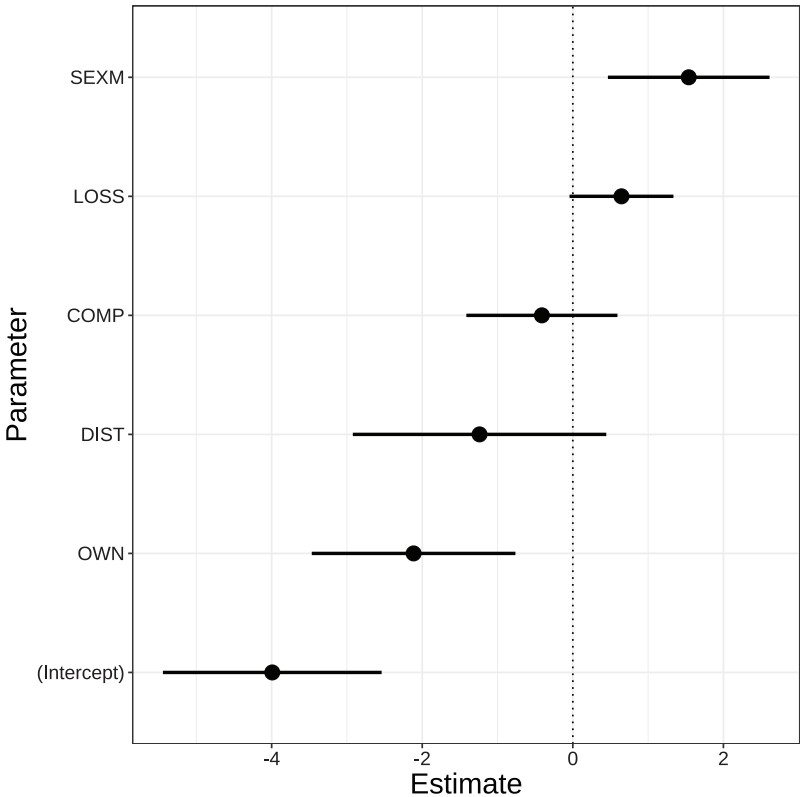

**Figure 3 Parameter estimates based on Generalized Linear Mixed-Effects Models of factors affecting perceptions to Himalayan wolves.** Dots and solid lines represent parameter estimates and 95% CI. The estimates are on the logit scale. The strength of the effect of parameters is indicated by the distances between the solid horizontal lines and the dotted vertical line. VDC (Village Development Committee) is included as a random effect. SEXM, Male; LOSS, Total livestock loss; COMP, Proportion of large livestock; DIST, Distance from the nearest conservation field office to respondent household; OWN, Total livestock holding.

*Lyngdoh & Habib, 2019*; *Suryawanshi et al., 2014*). The negative perceptions of the wolf in our study area is probably due to fear and cultural bias as reported in many other studies (*Linnell et al., 2002*; *Prokop, Usak & Erdogan, 2011*), an issue to be considered in future conservation plans in ACA and MCA.

In our study area, analysis of livestock depredation revealed higher losses from snow leopards compared to wolves (*Chetri et al., 2019a*), but still the tolerance level of local communities towards snow leopards was higher. Tolerance to snow leopards in ACA has changed to become more positive compared with an earlier study (*Oli, Taylor & Rogers, 1994*), probably as a result of continued efforts to increase awareness as part of an ongoing conservation program (*DNPWC, 2017*). No such efforts have targeted wolves, or other coexisting carnivores in this area.

Our model revealed that literacy, age, occupation, number of livestock owned and gender affected perceptions towards snow leopards and wolves. However, the predictors for the two species were different (see Fig. 2 and 3), that is, the latter two appeared in the best model for wolf perceptions and the former three for snow leopard. Regarding literacy and age, only the interaction between these terms had an influence on perceptions

of snow leopards, but not the main effects. Being illiterate was associated with negative perceptions among older respondents. Possibly, younger people had more exposure to snow leopard conservation campaigns, regardless of literacy. Several earlier studies have shown that older people are more negative towards large predators and usually less supportive of their conservation than the younger generation (*Bencin, Kioko & Kiffner, 2016*; *Kellert & Berry, 1987*; *Kleiven, Bjerke & Kaltenborn, 2004*; *Røskaft et al., 2007*; *Williams, Ericsson & Heberlein, 2002*).

Occupation influenced perceptions of snow leopards, that is, people with sources of income other than animal husbandry were more positive. Elsewhere, it was also reported that people having smaller landholdings and few economic opportunities other than livestock herding are more negative towards snow leopards and wolves (*Bagchi & Mishra, 2006*; *Din et al., 2017*). In a study of jaguars (*Panthera onca*), *Caruso et al. (2020)* found a similar pattern; people's perceptions and attitudes were strongly influenced by occupation and economic benefits through ecotourism. In Ladakh, India snow leopard based ecotourism has become popular and provides income generation opportunities to the local communities (*Jackson, 2015*; *Maheshwari & Sathyakumar, 2019*; *Vannelli et al., 2019*).

Regarding perceptions of wolves, males were more positive than females. This pattern was also reported in earlier studies (*Kellert & Berry, 1987*; *Røskaft et al., 2007*; *Suryawanshi et al., 2014*), and has been explained by women having less contact with conservation agencies (*Gillingham & Lee, 1999*). Another study suggested that the negative attitudes of women might be a result of greater perception of risk or fear (*Prokop & Fančovičová, 2010*). As suggested by *Kusi et al. (2019)*, men in the Himalayas often migrate outside of villages for seasonal work and may thus have been more exposed to alternative attitudes to nature and conservation. In addition, men frequently venture into the pasture for livestock grazing activities and presumably had more encounters with wolves, which make them understand their behavior and threats. High encounter rates with wolves either in the wild or in captivity, may promote more positive perceptions of the animals (*Arbieu et al., 2020*).

People holding large livestock herds were more negative towards wolves, which agrees with a study from western China (*Xu, Yang & Dou, 2015*). A possible explanation is that owners with larger herds have a higher risk of suffering losses in the central Himalayas (*Chetri, Odden & Wegge, 2017*). It is, however, notable that having experienced losses did not affect perceptions of snow leopards, and the effect of perception on wolves was weak. This is in contrast with a recent study from the Nepal Himalayas where livestock depredation by wolves is the main predictor of the negative attitude towards wolves (*Kusi et al., 2019*). However, such a pattern was not recorded in our study area and may be due to the fact that average losses in our study area were quite low (~1% of all livestock holdings).

In our study area, the NTNC has been implementing community-based conservation projects and programs since 1992. The overall goal is to conserve biodiversity of global significance with the active participation of local communities. Integrated conservation

and development efforts have therefore addressed the communities' needs and demands while actively mobilizing local people in conservation efforts. However, even after 2–3 decades of conservation initiatives, local perceptions and tolerance towards carnivores are still rather negative, particularly towards wolves. We therefore recommend a wider perspective of future awareness campaigns to include a broader specter of species and conservation issues with particular focus on the Himalayan wolf. During interviews, we observed that remote settlements had rarely been visited by conservation authorities, and the inhabitants there had limited knowledge of compensation policies for livestock losses and human injury. Local perceptions on wolves tended to be more negative with increasing distances from conservation field offices, and this has been reported in earlier research in parts of ACA (*Oli, Taylor & Rogers, 1994*). The factors underlying negative perceptions of distant settlements are probably due to a limited local involvement in community conservation programs. In the future, distant and remote settlements require more rigorous conservation outreach and awareness activities.

## CONCLUSIONS

This study has investigated local villagers' perception of snow leopards and wolves in the central Himalayas, Nepal. In general, the perceptions of locals were more positive towards snow leopards than to wolves. People having larger herds of livestock (goat/sheep) with limited access to conservation programs were more likely to have negative perceptions towards wolves. Our results showed that multiple factors influence local perceptions of the two carnivores and that perception factors cannot be generalized for the two species. Thus, they need to be addressed separately. We suggest that future conservation projects and programs prioritize remote settlements. Furthermore, considering the substantial influence of occupation on people's perceptions of carnivores, certain parts of the landscape, for example, Manang of ACA and Tsum valley of MCA, should be tested for the development of wildlife based ecotourism.

## ACKNOWLEDGEMENTS

We sincerely thank the survey participants, the field staff of ACA and MCA, and the local communities for their support and collaboration during the field surveys.

### Funding

Madhu Chetri's Ph.D. study was supported by the Norwegian State Education Loan Fund (Lånekassen) and Inland Norway University of Applied Sciences. The fieldwork was supported by the Panthera—the Kaplan Graduate Award. Fieldwork was also funded by the National Trust for Nature Conservation through its USAID funded Hariyo Ban Program. There was no additional external funding received for this study. The funders had no role in study design, data collection and analysis, decision to publish, or preparation of the manuscript.

## Grant Disclosures

The following grant information was disclosed by the authors:
Norwegian State Education Loan Fund (Lånekassen).
Inland Norway University of Applied Sciences.
Panthera—the Kaplan Graduate Award.
National Trust for Nature Conservation.

## Competing Interests

Thomas McCarthy is employed by Panthera. The authors declare that they have no competing interests.

## Author Contributions

- Madhu Chetri conceived and designed the experiments, performed the experiments, analyzed the data, prepared figures and/or tables, authored or reviewed drafts of the paper, and approved the final draft.
- Morten Odden conceived and designed the experiments, performed the experiments, analyzed the data, prepared figures and/or tables, authored or reviewed drafts of the paper, and approved the final draft.
- Olivier Devineau performed the experiments, analyzed the data, prepared figures and/or tables, authored or reviewed drafts of the paper, and approved the final draft.
- Thomas McCarthy conceived and designed the experiments, authored or reviewed drafts of the paper, and approved the final draft.
- Per Wegge conceived and designed the experiments, performed the experiments, analyzed the data, authored or reviewed drafts of the paper, and approved the final draft.

## Human Ethics

The following information was supplied relating to ethical approvals (i.e., approving body and any reference numbers):

The National Trust for Nature Conservation reviewed and approved this research and provided permission to carry out the study (Ref no. 291).

## Data Availability

Raw data are available in the Supplemental Files.

## Supplemental Information

Supplemental information for this article can be found online at http://dx.doi.org/10.7717/peerj.10108#supplemental-information.

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
