# Peer review of "Multiple factors influence local perceptions of snow leopards and Himalayan wolves in the central Himalayas, Nepal"

_PeerJ, doi:10.7717/peerj.10108_

## Round 0.1 · original submission · Major Revisions

Dear authors

Thanks for your interesting contribution.

Three reviews have been received and reviewers agree your manuscript is interesting but found major and minor issues to solve.

I will be glad to reconsider a resubmission if all comments performed by reviewers are taken into account and a detailed response letter to each point is received.

Best regards!

·

Basic reporting

The strength of this study is it includes relevant results for further research on the conservation strategies of these carnivore species in the central Himalayas. Moreover, the introduction and background are sufficient in terms of placing the reader in the study context. Regarding the format, the manuscript is correctly structured, fulfills the journal standards, and contains all the sections needed. Overall, the manuscript is clearly written in a professional language. However, several improvements with regard to basic reporting are required.

Major comments:
1. With regard to the Introduction and background of the study, a relevant point to explain here is that this research constitutes a further step in a long-term study (if not, explain why). I recommend including a justification of the novelty of the current study in relation to your previous studies in this matter (Chetri 2018).
2. Some grammar errors can be found along the manuscript and paragraph structure should be improved by polishing edition, particularly within the Introduction and Methods sections. Moreover, I highly encourage you to improve the paragraph structure in the Introduction section as follows: Lines 49-80 one paragraph is too long. I think that this aspect can be managed by creating two different paragraphs for each species information, focusing on local tolerance and ecological aspects (before Line 67), and then one more paragraph explaining common information (from Line 67 to 80). Furthermore, I suggest putting Lines 92-99 in another paragraph, as a specific statement of the questions formulated for this study. Lines 98-99, the size of the study area fits better in the Study area subsection.
3. I have some doubts relative to the questionnaire design because of certain questions formulated were excluded from the analyses, such as livestock numbers of each species (instead of the total number of domestic animals owned), season, losses experienced (yes/no) and reported, habitat type, etc. Why did you exclude these data from the analyses and results sections? Please, justify this and include a statement clarifying these omissions.

Minor comments:
1. (Line 62) Why the density estimates of wolves are still lacking? Is it because of the lack of recent DNA fecal study of wolves, or because this species is not easy to detect within the study area? How rare are the wolf species in the study area? Please, explain this and include this information if possible.
2. In general, in-text citations are well performed. Although they should be improved according to the Journal Instructions for Authors (check the Reference format section). A comma is required after the list of all author names and before the publication year, for example, Athreya et al., 2013. In line 41, remove the middle name of the second author (Cesar). In-Line 62, Chetri (2014) is not included in the Reference section. In line 306, add the publication year in the in-text citation. Moreover, the Reference Section’s guidelines recommend choosing an option to give the DOI on each reference. For example: DOI: 10.1016/j.jnc.2016.09.004. Please, apply these modifications along the manuscript.
3. According to PeerJ Ethic policies, an Ethics statement should be provided in the Materials and Methods section of the manuscript whenever the research was conducted on humans or human tissue; on animals or animal tissue. Please, include this statement related to Human participant's approval documentation obtained from the relevant approval body in the Material & Methods section (Line 147). If not being able to include it, please provide a statement explaining why it is not necessary.
4. In accordance with the PeerJ submission guidelines, all tables and figures titles should be in bold and with a colon after the figure or table name. Please check the format recommended and implement these changes along the manuscript but also in the figure and table files loaded. In more detail, Figure 1 requires more space at the map bottom to avoid cutting the map legend; the font of geographic coordinates of the map grid have to be increased to clear reading. If possible, try to reduce the white space inside the figure file loaded and cite map sources in the legend if you used anyone. In Figure 2, I recommend to define the VDC acronym in the legend but also to increase the font size in both axis titles and axis lines (and the same for Fig. 3). Table 1 can be improved as below: No. or number instead of No; “included or valid questionnaires” instead of “included respondents”; “invalid/excluded questionnaires” instead of “neutral responses”; all the units (i.e. years) should be placed in the row title in parentheses; all the acronyms and abbreviations defined in the footnote (VCD, ACA, MCA); all the footnote markers have to be referenced in the table; “proportions” should be changed by “percentage” or instead of that, specify what means here. Why are not included the answers relative to livestock losses and the loss numbers? Please, provide this information or explain why is not included. In Table 2, VDC would be defined in the legend (as well as in Table S1 and Table S2 in the Supplemental File), and selected models for each species highlighted in bold and this formatting explained in the legend. Moreover, you need to include the distance from the nearest conservation field office to the respondent household (DIST) as a predictor variable in the legend of Table 2 and Table S1, because it is included as a predictor variable in both cases.
5. Thank you for sharing the raw data and ensuring you follow our Data Sharing policy. However, supplemental files provided (peerj-48442-Raw_data_Snow_leopards.csv; peerj-48442-Raw_data_Himalayan_wolves.csv) require more descriptive column identifiers, including measurement units for each variable and acronyms or abbreviations definitions to easy reading. The file names provided in the “How are you submitting your raw data or code?” section do not correspond with the required ones. Please, correct it in this section.

Experimental design

The manuscript submitted is within the scope of this journal. The research aims sound rather clear and constitute relevant points for the study. In general, the research performance fulfills the basic technical and ethical standards (with the exception of minor comments made in the basic reporting). The methods section is rather detailed and contains relevant information to replicate the study. However, several improvements relative to the experimental design should be implemented, thus I would give some feedback relative to the Introduction and the Methods sections.

Major comments:
1. It is necessary to be more specific when defining the questions and justifying the need for this study in relation to the lack of knowledge about the problem analyzed (Lines 93-99). In Line 93, remove “in an area”.
2. I have found the questionnaire design used for data collection (Appendix S1) could be improved in order to answer the questions formulated. I believe that this issue can be solved if you justify the design provided in the Semi-structured questionnaire survey subsection and explain the reasons for excluding certain questions from the analyses in the data analysis subsection.
3. Why did you exclude the neutral respondents from the analyses? I believe that if you do not explain the exclusion reasons resembles a bias. Please, remove the neutral category from the answers because the response variable fits a binomial distribution (Line 155).
4. Change these two sentences (Lines 155-156) as follows: “We consider as invalid questionnaires when respondents did not want to answer or stated that they did not know about the species presence and conflict, thus they were excluded from the analyses” (see Appendix S1 comments). In Lines 158-159, change “neutral respondents” by “excluded or invalid questionnaires”.

Minor comments:
1. In Line 110, I believe that the preferred religion by the majority of the population is not relevant to the study. Please remove it, or instead of, please explain why the religion was not included as a factor in the analyses.
2. In the stud area subsection, there should be two different paragraphs between Lines 112-123 and Lines 123-128, or instead, try to summarize the information given in these lines.
3. In Lines 126-128, I have not found the link in this sentence to the rest of the paragraph, please remove it or create another paragraph to explain the distribution and ecological aspects of the study species.
4. In Line 167, are there any differences among the answers of individuals who have chosen the “others” option? It is possible to explain which answers were given as the “others” category and how did you manage this issue in the analyses.

Validity of the findings

I consider this study includes relevant findings for further research on the conservation strategies of these carnivore species in the central Himalayas. The statistical analyses seem well performed and the sample size used for the separated models is fairly balanced, and for both species, it allows a large number of factors and covariables to be included in model construction. Another good point is VDC was included as a random factor in both cases. Moreover, all the results were fully reported, and the figure design chosen to show the estimated effects is the most recommended. Most of the ideas argued in the Discussion section show a deep commitment to the labor of your team, and in general, most of the results are broadly supported. The conclusions highlight the most remarkable points of your study, but not all of them. To summarize, I believe the findings provided are rather robust and statistically sound, although certain clarifications in the Data analysis and the Results sections are required, as well as in the Discussion and the Conclusion sections remain open with respect to the background of the study.

Major comments:
1. The impact and the novelty of this research are not justified in the Introduction and Discussion sections. This manuscript will gain relevance if you improve these two aspects. For instance, which aspects of your research make a difference from what Kusi et al. (2019) observed?
2. I assumed that you performed a correlation test among predictor variables before the model construction; please confirm this doubt. In Line 28, how did you evaluate that the answer according to gender is independent of the number of losses? If possible, explain this point in the data analysis subsection. Moreover, I suggest you consider including the VDC as a nested factor of the number of losses variable.
3. According to Li et al. (2014), why did you not include religion as a predictive factor in both separated analyses? I suggest thinking about it particularly when you use this fact to give support in the Discussion section.

Minor comments:
1. Several points from the Discussion and the Conclusion sections remain open with respect to the background of the study. Between Lines 245-252, you suggested that all the commented cultural aspects are likely determining how have evolved local perceptions on snow leopards and Himalayan wolves. By contrast, you did not include religion as another factor in the analyses and you did not comment on the relevance of this aspect for the study species conservation in the introduction. Secondly, you suggested that the increasing tolerance for snow leopards could be linked with the continued efforts to increase awareness as part of the ongoing DNPWC conservation program (Line 266). However, you did not take into account for these measures in the data analysis and the introduction sections. Thirdly, as Caruso et al (2020) suggest (Lines 285-286), ecotourism development could be indirectly influencing local perceptions towards study species, though it was considered as the “other” category of occupation variable for the analyses. Lastly, the Oli et al. (1994) study reported that negative perceptions on wolves increased with the distances from conservation field offices, even so, you did not compare it when finding a weak effect of this variable (Lines 327-328). I encourage you to think about the ideas suggested and if possible, implement them in the new version of the manuscript.
2. The study concludes that the negative perceptions for wolves found in the study area are (Line 337: not is) probably due to fear and cultural bias as previously reported, although I also find interesting that this species suffers a scarcity in conservation awareness policies, and by contrast, the positive perception on snow leopards could be linked with higher conservation efforts. Apart from that, you have not controlled by this cultural bias in your analyses. Thus, I encourage you to implement these concluding remarks in the further version.

Additional comments

I really appreciate the study efforts during the data recording and the commitment of your team with the projects developed for both conservation species (ACA and MCA. I encourage you to improve this manuscript to provide the impact deserved of these study findings in order to promote further researches in conservation designing policies within the study area. I recommend you to apply the feedback provided in this review in order to get a new and improved version of the article for the second round of review.

Reviewer 2 ·

Basic reporting

I like to appreciate authors for conducting research in the world’s remotest area. It is a good research conducted focusing perception of people towards snow leopard and wolves. Definitely, the attitude of local people need to be understood for better conservation planning. Study has well speculated the perception of local people using generalized mixed effect model. However, the reasons behind the perception are not well discussed based on this study. I would like to suggest authors to focus rather on introduction and discussion while doing revision. I have pointed my comments line by line as follows.

Experimental design

No comment

Validity of the findings

Authors are suggested to support results with their own observation. I suggest authors to give backup from their own findings. This study should find out the reasons behind negative or positive perception relying on their studied area.

Additional comments

Line 39 – for better clearance write “Human-carnivore conflict”

Line 62 – Two sentence are contradicting each other. One sentence is saying density has not been estimated and another is saying rare. In these regions, very less study have been done. That does not mean species is rare.

Line 68/69 – Depredation mortality is rather low than other causes of death. Then, why livestock depredation is big issue?

Line 70/72 –Better to talk about prey killing behavior of wolves? Author should justify why killing is brutal?

It is suggested to authors to talk about the situation of retaliatory killing and livestock depredation in relation to perception.

Line 94-why livestock depredation is main concern? author should justify the statement somewhere in introduction.


Line 116 – Author is suggested to change “abundant flat plain” to “abundant pasture land”


Line 135/136 – VDCs are no more administrative unit in Nepal since 2015. VDC are changed to ward in most of cases and new rural municipalities have been established throughout the country. So, authors are suggested to review the study area accordingly.


Study area: Authors are suggested to touch briefly about human population and livestock numbers in study area. This will give readers idea about how human pressure and livestock pressure.

Line 159/160 – not clear. Authors are meant to say northwestern section of ACA and MCA or northwestern section of all studied settlements.

Line 162-172. In data analysis section; authors are suggested to justify “why your data fit to generalized mixed effect model” at the beginning and also suggested to include residual plot as supplementary information. As the model is mixed effect, it is good idea to plot by partially pooling. This will help to show relation with each group.


Line 174/175 –These two lines rather fit in methodology.

Discussion

Line 236-243 –Authors are suggested to support results with their own observation. Here in these lines, authors have supported their observation with other research. In discussion section, I found that findings of this research are justified by other research, for example in Line 252 to 260. I suggest authors to give backup from their own study. This study should find out the reasons behind negative or positive perception based on their studied area.

Line 247/249 – I think that traditional practice of grazing does not fit here. How are you relating this statement with your research finding?

Line 251/252 – “Buddhist community did not kill wildlife”. It seems that they do kill wildlife now. Authors are suggested to change the structure of sentence.

Line 275/276 – As authors have studied perception and age. They are suggested to compare age vs perception to conclude in different perception of young and old people. As you have mentioned about religious belief. Possibly, old people are more religious and should be more positive towards snow leopard. You have also mentioned that the conservation practice in the region is quite old. I think that that program must have been focusing entire population (young people, old people, both male and female) of your study area without any bias


Line 289-292 – these sentences are suitable for conclusion or in concluding part of discussion.

Line 305/306 – Incomplete citation


Conclusion
Line 332 to 342: authors are suggested to include only their findings in conclusion.

Annotated reviews are not available for download in order to protect the identity of reviewers who chose to remain anonymous.

Reviewer 3 ·

Basic reporting

no comment

Experimental design

The article is framed within the social sciences but with clear applicability in environmental sciences.

Given the scope of the journal I would recommend the authors to make brief recommendations for the conservation (or to foster tolerance) of these two species at the end of the manuscript

Methods are well described in general but I have doubts in the definition of the variable COMP (proportion of large stock). I interpret with this that only certain species (large ones) are considered for this variable, if so, which are they?

Validity of the findings

no comment

Additional comments

I have enjoyed reading this article which deals with the local perceptions of two large carnivores in the central Himalayas. As the authors said in the abstract " An understanding of local perceptions of carnivores is important for conservation and management planning" However, I believe that the article may benefit if the authors connect these perceptions with management planning by including a series of recommendations based on these results that may in some way favor co-existence in the area of study. I understand that they include a few sentences during the discussion but I think it would be useful to address this in further detail.

I don't know where it would fit better (introduction or discussion) but I think it would be good to include a small paragraph about the benefits (apex predator, ecotourism opportunities...) that these carnivores can bring in the area and not only focus on them as livestock predators.

I'm sorry, possibly I have not understood well, throughout the text you mention two areas ACA and MCA. I understand that they differ in the type of species of livestock (among others characteristics I guess) but then I do not see that the analysis takes into account whether the respondents are from one area or another. Would it not be appropriate to include the regions in the analysis or the type of livestock they had?

Appendix S1 Questionnaire:
-Is there any relationship between having an encounter with the species (question 6 of the questionnaire) and the type of perception (positive/negative)?
-It would be interesting for the reader to have some more descriptive results from the questionnaire (for example questions 3, 4, 6 and 10)
-Questions 8 and 9: You ask why but I find no mention of these questions in the manuscript. Only just a suggestion, I find it interesting if the authors decide to analyze these questions. For example, a content analysis would be a good tool to determine which words could be associated with positive/negative perceptions of species.

I have other minor suggestions:

L31 future conservation projectS
L 64 ACA this acronym is not yet defined in the text
L66 Do you have any knowledge about the distribution ranges of these species that could be projected on the map?
L107 What kind of tourism? (mountaineering, or wildlife watching) Are there eco-tourism options for these species in the area?
L137 do you know how these VDC (or differences between ACA and MCA) manage the attacks of the carnivores (legal hunting, lethal control, fences, aversive conditions...)
L236 Typo: This perception (without s)
L300 I think it should be "may have been"
L 304 some mistake by the reference manager.
L309 I think that “actually” it is not necessary here
L312 delete comma
L315-316 so why did you do not ask it in the questionnaire. I’m wondering if it also could be related with tolerance to snow leopard and wolves.
L327 “ perceptions of distant settlements ARE due to THE limited nature…”
L315-216 it would have been a good idea to include in the questionnaire some questions related to people's participation in conservation tasks
L335 “…perceptions for THE Wolf in our study area ARE probably…”

---

## Round 0.2 · Minor Revisions

Dear authors, one of the previous reviewers has kindly rereviewed your manuscript and has agreed that it is very much improved. The reviewer found minor issues with the text that should be fixed.

Please carefully implement the changes suggested by the reviewer so the manuscript reaches its best potential before publication.

Best regards

·

Basic reporting

The manuscript fulfills the general standards and all improvements suggested were effectively performed. I consider the introduction to have improved notably, and the background and the aims of the study are now sufficient to justify the novelty of the current research. The polishing edition is notable in the new version of the manuscript. However, some minor comments are included below for improving it.

Minor comments:

1. Thank you for including all the necessary corrections regarding the in-text citations and references section. In Line 42, please remove “de” before “Azevedo”. In Line 55, please remove the second dot in the last citation. In Line 82, I would suggest adding the reference citation (Chetri et al., 2019a), as an example of the background of the study. Please notice that parentheses () are not required after “et al.” in Line 194. In Lines 200, 252, and 304, a comma (,) is not required after the author names. In Line 512, Mijiddorj, Alexander and Samelius (2018) reference is not cited in the text. In Line 577, Zimmermann, Wabakken and Dötterer (2001) reference should be in a separate line. According to PeerJ submission guidelines, for three or fewer authors, you need to list all author names (e.g. Zimmermann, Wabakken and Dötterer, 2001). Please check the references section and in-text citations.

2. I appreciate all the improvements implemented in the introduction section. However, I believe that the question concerning the density of Himalayan wolves has not been resolved yet. I would suggest taking into account the following references “WWF (2015) Non-Invasive Genetic Population Survey of Snow Leopard and Wolf. Final report. WWF, Kathmandu, Nepal”; “Chetri et al. (2016) ZooKeys; and Chetri et al. (2017) PLoS ONE” and include them in Line 72. In Line 75, please consider removing “However” in the beginning, there is no opposition here. Thank you for defining the acronyms in Lines 86-87. I would suggest adding the following specification: (ACA, hereafter) and (MCA, hereafter). In Line 101, I would recommend implementing the plural in “low livestock loss”.

3. I believe that the statement in the data analysis subsection helps to clarify the selection of the most appropriate predictors recorded through the semi-structured questionnaire. Thank you for including all the recommended predictors in the analysis, and all the changes needed in all tables and figures. Please consider including units (years) for “Mean age” and correct “parentheses” in the footnote of Table 1. Moreover, I recommend to add the definition of ownership predictor, and to replace “LIT1” with “LITyes”, and “AGE:LIT1” with “AGE:LITyES” in the legend of Figure 2 (please, check if the figure shown matches the selected model). Please, consider replacing “snow leopard” with “Himalayan wolf” when defining the LOSS predictor in the footnote of Table S2 (Supplemental files).

Experimental design

The study aims are well defined in the introduction section, and the novelty of the current research is well justified. Moreover, the research complies with technical and ethical requirements, and the methods section shows significant improvements, according to the comments received. Thank you for removing neutral respondents and for including a statement in the data analyses subsection in order to clarify the study design and model selection. However, I will provide some feedback in order to improve the manuscript.

Minor comments:

1. Thank you for improving the paragraph structure in the study area subsection. In Line 137, I recommend changing “Bos spp.” to the italic font.

2. In Line 176, I understand that the invalid questionnaires belong to respondents who gave neutral answers or ignored the presence of study species or conflicts with them. If that is correct, I would suggest including “(i.e., neutral responses)” after "conflict" for further clarification.

3. I agree with the reasons for excluding religion from the analyses. However, I would consider this explanation (or similar) to be included in Line 199: “we did not include religion as a predictive factor as all the locals residing in the region are Buddhist, only a few outsiders who are working as labor or teacher are non-Buddhist.”

Validity of the findings

The results of this study are widely supported. The findings are quite robust and statistically sound, with a significant application to future conservation plans for these large carnivore species in the Himalayan range. I would highlight the improvements made in the introduction as well as in the discussion and conclusion sections, which are now further strengthened in terms of the aims and background of this study. Thank you for including all the comments provided previously, particularly with reference to statistical analysis. However, I will provide some feedback to improve your manuscript.

Minor comments:

1. Results are rather strong and robust. However, I would suggest replacing “literate” with “being illiterate” in Line 231. Moreover, I would recommend checking if Figure 2 matches with the selected model. In this case, please consider adding "and ownership" after “sex” in Line 237.
2. Thank you for all your answers and changes implemented in the discussion section. In Line 266, I recommend adding “, to which most of our respondents belong,” after “Buddhist communities”. In Line 275, I would suggest replacing “dislike for” with “aversion to”. Please consider replacing “An other” with “another” in Line 313, and “perceived” with “perception of” in Line 314. In Line 353, please leave a space between “Theyneed”.

3. With regard to the impact of your findings, I suggest remarking which aspects are different from the study of Kusi et al. (2019). For example, your research was performed in a different study area. In Line 329, I would recommend comparing your results here in more detail (Fig. 3) in contrast to Kusi et al.

4. Thank you for emphasizing the difference in perceptions of wolves and snow leopards in the introduction and the discussion sections. However, I recommend adding “an issue to be considered in future conservation plans in ACA and MCA” at the end of Line 279. In Line 339, after “conservation issues”, please consider adding “with particular efforts to the Himalayan wolf”.

Additional comments

I really appreciate all the improvements implemented in the current version of the manuscript. I would highlight the research efforts and the commitment of your research group with the projects developed for both conservation species in ACA and MCA. I highly encourage you to improve this manuscript and give these results the impact they deserve, with the ultimate goal of getting more funding for monitoring these two carnivore species and promoting future conservation and management strategies in remote settlements, particularly focused on the Himalayan wolf. I recommend you to apply the feedback provided in this review in order to get a more polished version of the article before publication.

---

## Round 0.3 · accepted · Accept

Dear Authors,

Congratulations on your interesting work. Now accepted for publication in PeerJ.